# Sporadic Spinal-Onset Amyotrophic Lateral Sclerosis Associated with Myopathy in Three Unrelated Portuguese Patients

**DOI:** 10.3390/brainsci13020220

**Published:** 2023-01-28

**Authors:** Miguel Oliveira Santos, Marta Gromicho, Ana Pronto-Laborinho, Mamede de Carvalho

**Affiliations:** 1Department of Neurosciences and Mental Health, Hospital de Santa Maria, Centro Hospitalar Universitário de Lisboa Norte, 1649-028 Lisbon, Portugal; 2Instituto de Fisiologia, Instituto de Medicina Molecular, Centro de Estudos Egas Moniz, Faculdade de Medicina, Universidade de Lisboa, 1648-028 Lisbon, Portugal

**Keywords:** amyotrophic lateral sclerosis, myopathy, muscle fibre lesion, myopathy, pathogenesis, progression

## Abstract

Amyotrophic lateral sclerosis (ALS) and myopathy have been already described as part of a common genetic syndrome called multisystem proteinopathy. They may occur together or not, and can be associated with other clinical features such as frontotemporal dementia and Paget’s bone disease. In addition, primary skeletal muscle involvement has been also reported in inherited forms of lower motor neuron disease, in spinal–bulbar muscular atrophy and in spinal muscular atrophy. We aim to characterize three sporadic, spinal-onset ALS patients, one with a concurrent non-specific myopathy, and two with a previous diagnosis of myopathy before upper and lower motor neuron signs emerged. Perhaps our sporadic ALS cases associated with myopathy share a common, but still unknown, pathogenic background. These cases raise the paradigm of a possible interplay between skeletal muscle degeneration and motor neuron damage.

## 1. Introduction

Amyotrophic lateral sclerosis (ALS) is a rapidly progressive and fatal motor neuron disorder characterized by degeneration of the upper (UMNs) and lower motor neurons (LMNs) [1]. While the disease is inherited in about 5–10% of cases, most cases are classified as sporadic [1]. *C9orf72* gene mutations represent the most common genetic abnormality both in familial (≈40%) and sporadic (≈5–7%) cases of ALS, and in frontotemporal dementia (≈20–40%) [2]. Other mutations are much less common [1].

Up to now, five genes (*VCP, SQSTM1*, *HNRNPA2B1*, *HNRNPA1* and *MATR3)* have been associated with multisystem proteinopathy syndrome, which usually involve muscle and additional organs/systems, including UMNs and LMNs [3,4,5,6]. In both spinal muscular atrophy (SMA) and spinal–bulbar muscular atrophy (SBMA or Kennedy’s disease), inherited forms of LMN disease, an associated skeletal muscle involvement has been described [7,8,9,10]. 

We aim to report and discuss three sporadic spinal-onset ALS patients, one with concurrent neurophysiological and histopathological features of myopathy, and two with a previous diagnosis of a myopathy before the clinical signs of motor neuron loss emerged.

## 2. Patient 1

A 61-year-old Portuguese woman presented with progressive disto-proximal right lower limb muscle weakness at the age of 59. After two months, she complained about the same pattern of muscle weakness on the contralateral side and started to use a cane to walk. She lost ambulation and became wheelchair bound six months later. At the age of 60, bilateral hand muscle weakness emerged. A progressive dysarthria started one year later. There were no dysphagia, respiratory or sensory symptoms, and her weight was stable. Both past medical and family history were unremarkable. At the age of 61, neurological examination revealed symmetric paresis and atrophy of the arms (biceps and deltoid = 4 and hand muscles = 2 on the MRC scale) and legs (psoas = 0, quadriceps and abductors = 2, tibialis anterior and gastrocnemius = 0 on the MRC scale), neck weakness (flexion = 3 on the MRC scale), and mildly atrophic tongue, but fasciculations were not observed in any muscle. Jaw, upper limbs and patellar reflexes were very brisk. The Hoffman sign was present on the right side, but the plantar response was absent bilaterally. Cough reflex was weak. Spasticity and sensory changes were absent, and cognition was normal. 

Blood tests were unremarkable, including sedimentation rate, serum creatine kinase (CK) and autoimmunity screening (anti-nuclear, anti-double stranded DNA, ANA screening, lupus anticoagulant, anti-cardiolipin, anti-β2-glycoprotein-I, anti-Jo-1, anti-Mi-2 and anti-SRP antibodies). Serology for HIV 1/2, hepatitis B/C, Lyme and syphilis were negative. CSF analysis was unrevealing, including neurotropic virus panel and anti-neuronal antibodies (anti-Hu, anti-Yo, anti-Ri, anti-CV2, anti-SOX1 antibodies). Sensory nerves conduction studies were normal. Motor responses were absent (lower limbs) or of small amplitude (upper limbs), but conduction velocities were normal in the forearms bilaterally. Needle sampling showed abnormal spontaneous activity (fibrillation potentials/positive sharp waves) and mainly myopathic potentials (short duration, small amplitude and polyphasic) in biceps brachii, first dorsal interosseous, vastus medialis and tibialis anterior, bilaterally. However, in these muscles some large amplitude motor unit potentials with increased firing rate were observed. Sternocleidomastoid and genioglossus muscles were considered normal. Brain and spinal cord MRI were normal. Muscle MRI of the lower limbs showed symmetrical signs of fatty degeneration, with major signal increase in fat suppression sequences (Figure 1). Right deltoid biopsy disclosed mixed myopathic (type 1 fibre predominance) and neurogenic (type-grouping) findings. There were no rimmed vacuoles, cytoplasmic inclusions or inflammatory infiltrate. Dystrophin (Dys 1, Dys 2 and Dys3 fractions), sarcoglycans (α, β, δ and γ) and laminin M-chain (merosin) expression was normal. Sarcolemma MHC class I staining was normal. One year later, a clear neurogenic pattern was also disclosed in sternocleidomastoid and genioglossus muscles, with large motor units, complex and unstable, with increased firing rate. 

A tentative diagnosis of probable laboratory-supported ALS was made according to the revised El Escorial Criteria and Awaji guidelines [11,12]. A coexistent non-specific myopathy was accepted. She initiated riluzole 50 mg b.i.d. A screening for muscle dystrophy genes was negative. Whole-exome sequencing (WES) was negative for all the analysed genes, including those associated with multisystem proteinopathy. Over the next 2–3 years she progressed to respiratory insufficiency and bulbar dysfunction with severe dysphagia and ultimately died.

## 3. Patient 2

A 76-year-old Portuguese woman presented with progressive proximal upper and lower limb muscle weakness, bilaterally, at the age of 60. There were no bulbar or sensory symptoms. Both past medical and family history were unremarkable. On examination it was noticed a symmetrical limb-girdle weakness pattern (biceps and deltoid = 4 and psoas = 4 on the MRC scale). Ocular, facial, bulbar and neck muscles were spared. Patellar reflexes were brisk, but plantar response was flexor, bilaterally. There were no fasciculations, spasticity or calf pseudo-hypertrophy. Sensation, respiration and cognition were also normal.

Blood tests were unremarkable, except for increased CK (1021 UI/L, normal <180). The autoimmune screening panel (see above) was negative. Biceps and vastus medialis electromyography showed profuse fibrillation potentials/positive sharp waves, associated with a myopathic pattern suggesting an inflammatory myopathy. Right deltoid muscle biopsy disclosed severe fibre atrophy and focal necrotic fibres without significant endomysial inflammatory infiltrate. Immunohistochemistry reactions revealed T lymphocytes (positive for CD4), and macrophages (positive for CD68) near the necrotic fibres. Sarcolemma MHC class I staining was positive. These findings were consistent with a necrotizing autoimmune myopathy. Gastrointestinal endoscopies and whole-body CT scan did not find a tumour. She improved on prednisolone (1 mg/kg/day) and methotrexate (15 mg/week). Later, intravenous immunoglobulin was associated to ensure remission.

At the age of 74 she presented rapid progression of weakness in the arms and legs, associated with abnormal brisk upper and lower limb reflexes, and weight loss, in spite of a higher dose of steroids and intravenous immunoglobulin. Electromyography revealed fibrillation potentials/positive sharp waves and chronic neurogenic changes in the bulbar, cervical, dorsal and lumbosacral regions, with normal sensory and motor conduction studies. A diagnosis of probable laboratory-supported ALS was made according to the revised El Escorial and Awaji Criteria [11,12]. She started riluzole 50 mg 2id. In the follow-up for the next 6 years, she developed progressive bulbar dysfunction (dysarthria, dysphagia, tongue atrophy with fasciculations), very weak cough reflex, weak neck muscles, widespread fasciculations in the upper and lower extremities, severe distal limb weakness, and severe respiratory involvement (non-invasive ventilation >22 h per day). Eventually she died from respiratory complications. 

## 4. Patient 3

A 68-year-old Portuguese woman presented with slowly progressive proximal muscle weakness, without bulbar, ocular, cognitive or respiratory symptoms. Her neurologic examination revealed symmetrical moderate (MRC grade 4) weakness of shoulder and hip girdle muscles, with normal deep tendon reflexes, plantar responses, and sensation, and no fasciculations, atrophy or spasticity. CK level was mildly elevated (374 UI/L, normal <180), nerve conduction studies were normal but needle EMG disclosed myopathic potentials in the proximal arm and leg muscles. Electrocardiogram and echocardiography investigations were normal. Muscle biopsy of the right deltoid revealed increased fibrosis, with few fibres with amorphous hyaline deposits (modified Gomori trichrome stain) devoid of oxidative enzyme activity (NADH dehydrogenase stain) and cytoplasmic bodies (modified Gomori trichrome stain). The diagnosis of myofibrillar myopathy was established. The patient declined genetic testing for myofibrillar myopathy.

She decided not being followed by a neurologist for the next 8 years due to minor clinical progression. At the age of 77 she returned to the neuromuscular clinic due to rapidly severe gait impairment, requiring a walking frame to ambulate. On examination we observed a marked lower limb spasticity more severe on the right, with brisk upper and lower limbs deep tendon reflexes, brisk jaw reflex, Hoffman sign bilaterally, extensor plantar response on the right side, widespread fasciculations in proximal limb muscles, and dysarthria. Sensory examination, respiration, coordination, and cognition were normal. Brain and cervical cord MRI were unremarkable. Nerve conduction studies were normal, but needle EMG disclosed fibrillation and sharp waves, fasciculation potentials, poor interferential pattern on full contraction and neurogenic motor units in proximal and distal muscles of the arms and legs. A diagnosis of probable ALS was made supported by the revised El Escorial Criteria and Awaji guidelines [11,12]. She persisted in declining genetic testing. She was followed for approximately one year, presenting fast progression to anarthria, dysphagia and respiratory insufficiency, with tongue atrophy showing fasciculations, diffuse muscle wasting and very severe functional impairment; eventually she died.

## 5. Discussion

We report three sporadic, spinal-onset ALS patients, one with a concurrent non-specific myopathy, and two with a previous diagnosis of myopathy before rapid clinical deterioration associated with both UMN and LMN signs emerged.

Motor neuron loss and primary skeletal muscle lesion have been increasing recognized in some complex neurological disorders, including multisystem proteinopathy and inherited forms of LMN disease.

Multisystem proteinopathy is a rare, late-onset, progressive autosomal dominant disorder commonly caused by heterozygous missense mutations in the *VCP* gene [13,14,15]. Mutations in other genes, such as *SQSTM1*, *HNRNPA2B1*, *HNRNPA1*, and more recently *MATR3*, have been also described [4,5,6,16,17]. Clinical presentations usually include a variable association of proximal or distal inclusion body myopathy (≈90%), Paget’s disease of bone (≈50%), frontotemporal dementia (≈30%) and ALS (≈10%) [3,18]. Phenotypical heterogeneity is well known even within families [19,20]. A mixed neurogenic/myopathic or, less frequently, a purely neurogenic or myopathic pattern may be disclosed on electromyography [21]. Muscle histopathology may reveal rimmed vacuoles and ubiquitin-, TDP-43- and VCP-positive cytoplasmic inclusions [3,15]. Nonetheless, coexistent neurogenic findings may also be observed [3]. 

SMA has traditionally been classified as a pure LMN disease, but it is becoming increasingly clear that other organs, including the skeletal muscle, may be primarily affected due to reduced levels of SMN1 [7,8]. Actually, autophagy in SMA muscle seems to be modified in myotube atrophy [8]. Moreover, polyglutamine expansions in the androgen receptor gene causing spinal and bulbar muscular atrophy originates with muscle fibre dysfunction in a mouse model, before changes in the spinal cord are observed [9,22]. 

WES in our first patient did not reveal any possible known genetic background for both ALS or myopathy, but some aspects may look like multisystem proteinopathy, including the late-onset presentation, and simultaneous involvement of motor neurons and skeletal muscles confirmed by both EMG and muscle biopsy. However, the absence of vacuoles and cytoplasmic protein aggregates on the muscle biopsy is against this hypothesis. Although no pathogenic mutation was found by WES, we cannot rule out intronic mutations or mutations in still unknown genes. We suggest that this ALS-myopathy case have a common pathophysiological background with a still unidentified genetic abnormality. The other two patients had a previous diagnosis of a specific myopathy, inflammatory and myofibrillar myopathy respectively, before the emergence of UMN and LMN signs, and a possible interplay between skeletal muscle degeneration and motor neuron damage may be suggested.

Although we cannot rule out that it could be coincidental disorders, their rarity and in the light of what is known regarding other multifaceted neurological disorders involving both motor neurons and skeletal muscles, reinforce the emerging concept of a muscle role in causing motor neuron dysfunction [23]. Further studies should approach this exciting challenge.

## Figures and Tables

**Figure 1 brainsci-13-00220-f001:**
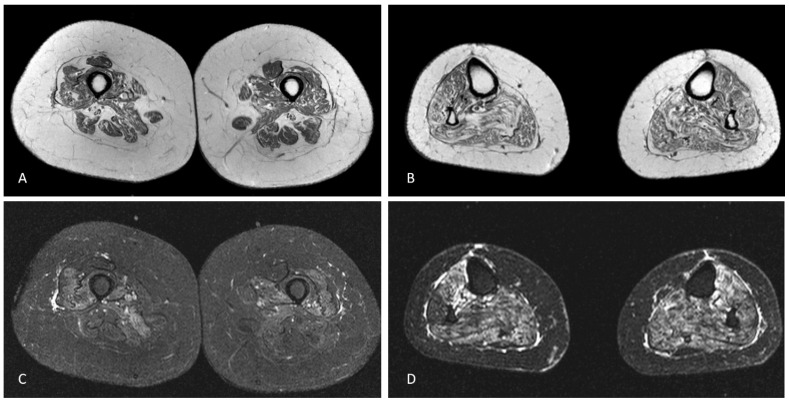
Axial T1-weighted MRI of the mid-thighs (**A**) and middle third of the legs (**B**) demonstrating mild, symmetric muscular fatty atrophy of the quadriceps muscle and the hamstring muscles and moderate fatty atrophy of all leg muscles. Axial STIR-weighted images of the thighs (**C**) and legs (**D**) at the same levels of A and B showing mild oedema of vastus medialis, vastus intermedius, vastus lateralis and, to a lesser degree, of the hamstring muscles in the thighs and severe oedema of all leg muscles.

## Data Availability

Data will be share upon reasonable request.

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
