# Peer review of "Sporadic Spinal-Onset Amyotrophic Lateral Sclerosis Associated with Myopathy in Three Unrelated Portuguese Patients"

_brainsci, 2023, doi:10.3390/brainsci13020220_

Round 1

Reviewer 1 Report

The paper is a compilation of 3 case studies describing ALS in patients of Portuguese descent.  The unique feature associated with these case studies is the concurrent myopathy.  I am not a clinician, so I can speak well to the clinical assessment.   The authors point to a possible underlying genetic cause and although they did perform exome sequencing in just the one patient, they very unfortunately did not embark upon whole genome sequencing for all three patients, which would have greatly strengthened this paper.  Its really unfortunate that sequencing was not performed.  I also thought the discussion was not very complete and could use improvement.  There were vague references to SMA at the end that were awkward.

Author Response

Dear Editor and Reviewers,

We would like to thank the Editor and Reviewers for all the observations and questions raised regarding the submission of our manuscript. We appreciated much that a revised version of our manuscript might be reconsidered. Below we present all the answers to the raised observations and questions. We have also done all the pertinent changes in the text itself, as suggested by the Editor and Reviewers.

R: The authors point to a possible underlying genetic cause and although they did perform exome sequencing in just the one patient, they very unfortunately did not embark upon whole genome sequencing for all three patients, which would have greatly strengthened this paper. It’s unfortunate that sequencing was not performed.

We would like to thank the reviewer for this comment. In the first patient, we have anticipated a possible undefined common genetic cause due to a concurrent ALS-myopathy case. For this reason whole exome was done but whole genome, this is a limitations we have added in discussion.  In another case we diagnosed necrotizing myositis, a non-genetically related condition. Finally, regarding the third patient, her myofibrillar myopathy emerged before clinical evidence of motor neuron disease, and this patients declined genetic testing.

 I also thought the discussion was not very complete and could use improvement. There were vague references to SMA at the end that were awkward.

We report three sporadic, spinal-onset ALS patients, one with a concurrent non-specific myopathy, and two with a previous diagnosis of myopathy before rapid clinical deterioration associated with both UMN and LMN signs emerged. The reference to multisystem proteinopathy and SMA aim to share the increasing evidence of known neurological disorders in which skeletal muscle (primary) involvement and motor neuron loss may coexist: “Motor neuron loss and primary skeletal muscle lesion have been increasing recognized in some complex neurological disorders, including multisystem proteinopathy and inherited forms of LMN disease.” We have also added in the discussion: “WES in our first patient did not reveal any possible known genetic background for both ALS-myopathy, but some aspects may look like multisystem proteinopathy, including the late-onset presentation, simultaneous involvement of motor neuron and skeletal muscle confirmed by both EMG and muscle biopsy. However, the absence of vacuoles and cytoplasmic protein aggregates on the muscle biopsy is against this hypothesis. Although no pathogenic mutation was found by WES, we cannot rule out intronic mutations or mutation in still unknown genes. We suggest that this ALS-myopathy case have a common pathophysiological background with a still unidentified genetic. Abnormality. The other two patients had a previous diagnosis of a specific myopathy, inflammatory and myofibrillar myopathy respectively, before the emergence of UMN and LMN signs, and a possible interplay between skeletal muscle degeneration and motor neurons damage may be suggested.

Although we cannot rule out that it could be coincidental disorders, their rarity and in the light of what is known regarding other multifaceted neurological disorders involving both motor neuron and skeletal muscle, reinforce the emerging concept of a muscle role in causing motor neuron dysfunction.23 Further studies should approach this exciting challenge..”

Reviewer 2 Report

The cases are well presented. At the same time the relation between the cases and multisystem proteinopathy is not persuasive.

The genetic analysis of the first case didn't reveal any  possible genetic background for the multisystem proteinopathy.

The last two cases seem more the overlapping of ALS on the picture of myopathy.

In spite that lots of data are presented for the cases the discussion is vague, lot of more arguments for the atribution of the mentioned cases to the multisystem proteinopathy should be given. 

Author Response

Dear Editor and Reviewers,

We would like to thank the Editor and Reviewers for all the observations and questions raised regarding the submission of our manuscript. We appreciated much that a revised version of our manuscript might be reconsidered. Below we present all the answers to the raised observations and questions. We have also done all the pertinent changes in the text itself, as suggested by the Editor and Reviewers.

The cases are well presented. At the same time the relation between the cases and multisystem proteinopathy is not persuasive. The genetic analysis of the first case didn't reveal any possible genetic background for the multisystem proteinopathy. The last two cases seem more the overlapping of ALS on the picture of myopathy. In spite that lots of data are presented for the cases the discussion is vague, lot of more arguments for the attribution of the mentioned cases to the multisystem proteinopathy should be given.

We report three sporadic, spinal-onset ALS patients, one with a concurrent non-specific myopathy, and two with a previous diagnosis of myopathy before rapid clinical deterioration associated with both UMN and LMN signs emerged. The reference to multisystem proteinopathy and SMA aim to share the increasing evidence of known neurological disorders in which skeletal muscle (primary) involvement and motor neuron loss may coexist: “Motor neuron loss and primary skeletal muscle lesion have been increasing recognized in some complex neurological disorders, including multisystem proteinopathy and inherited forms of LMN disease.” We have also added in the discussion: “WES in our first patient did not reveal any possible known genetic background for both ALS-myopathy, but some aspects may look like multisystem proteinopathy, including the late-onset presentation, simultaneous involvement of motor neuron and skeletal muscle confirmed by both EMG and muscle biopsy. However, the absence of vacuoles and cytoplasmic protein aggregates on the muscle biopsy is against this hypothesis. Although no pathogenic mutation was found by WES, we cannot rule out intronic mutations or mutation in still unknown genes. We suggest that this ALS-myopathy case have a common pathophysiological background with a still unidentified genetic. Abnormality. The other two patients had a previous diagnosis of a specific myopathy, inflammatory and myofibrillar myopathy respectively, before the emergence of UMN and LMN signs, and a possible interplay between skeletal muscle degeneration and motor neurons damage may be suggested.

Although we cannot rule out that it could be coincidental disorders, their rarity and in the light of what is known regarding other multifaceted neurological disorders involving both motor neuron and skeletal muscle, reinforce the emerging concept of a muscle role in causing motor neuron dysfunction.23 Further studies should approach this exciting challenge.”

Round 2

Reviewer 2 Report

Thank you for the provided explanations.